# Frequency of Androgen Receptor Positivity in Tumors: A Study Evaluating More Than 18,000 Tumors

**DOI:** 10.3390/biomedicines12050957

**Published:** 2024-04-25

**Authors:** Florian Viehweger, Jennifer Hoop, Lisa-Marie Tinger, Christian Bernreuther, Franziska Büscheck, Till S. Clauditz, Andrea Hinsch, Frank Jacobsen, Andreas M. Luebke, Stefan Steurer, Claudia Hube-Magg, Martina Kluth, Andreas H. Marx, Till Krech, Patrick Lebok, Christoph Fraune, Eike Burandt, Guido Sauter, Ronald Simon, Sarah Minner

**Affiliations:** 1Institute of Pathology, University Medical Center Hamburg-Eppendorf, 20246 Hamburg, Germany; f.viehweger@uke.de (F.V.); j.hoop@gmx.net (J.H.); c.bernreuther@uke.de (C.B.); f.buescheck@uke.de (F.B.); t.clauditz@uke.de (T.S.C.); a.hinsch@uke.de (A.H.); f.jacobsen@uke.de (F.J.); luebke@uke.de (A.M.L.); s.steuer@uke.de (S.S.); c.hube@uke.de (C.H.-M.); m.kluth@uke.de (M.K.); t.krech@uke.de (T.K.); p.lebok@uke.de (P.L.); c.fraune@uke.de (C.F.); e.burandt@uke.de (E.B.); g.sauter@uke.de (G.S.); s.minner@uke.de (S.M.); 2Pathologie-Hamburg, Labor Lademannbogen Medizinisches Versorgungszentrum (MVZ) GmbH, 22339 Hamburg, Germany; 3Department of Pathology, Academic Hospital Fuerth, 90766 Fuerth, Germany; andreas.marx@klinikum-fuerth.de; 4Institute of Pathology, Clinical Center Osnabrueck, 49076 Osnabrueck, Germany

**Keywords:** androgen receptor, immunohistochemistry, cancer, tissue microarray, prognosis

## Abstract

Androgen receptor (AR) is a transcription factor expressed in various normal tissues and is a therapeutic target for prostate and possibly other cancers. A TMA containing 18,234 samples from 141 different tumor types/subtypes and 608 samples of 76 different normal tissue types was analyzed by immunohistochemistry. AR positivity was found in 116 tumor types including 66 tumor types (46.8%) with ≥1 strongly positive tumor. Moderate/strong AR positivity was detected in testicular sex cord-stromal tumors (93.3–100%) and neoplasms of the prostate (79.3–98.7%), breast (25.0–75.5%), other gynecological tumors (0.9–100%), kidney (5.0–44.1%), and urinary bladder (5.4–24.2%). Low AR staining was associated with advanced tumor stage (pTa versus pT2-4; *p* < 0.0001) in urothelial carcinoma; advanced pT (*p* < 0.0001), high tumor grade (*p* < 0.0001), nodal metastasis (*p* < 0.0001), and reduced survival (*p* = 0.0024) in invasive breast carcinoma; high pT (*p* < 0.0001) and grade (*p* < 0.0001) in clear cell renal cell carcinoma (RCC); and high pT (*p* = 0.0055) as well as high grade (*p* < 0.05) in papillary RCC. AR staining was unrelated to histopathological/clinical features in 157 endometrial carcinomas and in 221 ovarian carcinomas. Our data suggest a limited role of AR immunohistochemistry for tumor distinction and a prognostic role in breast and clear cell RCC and highlight tumor entities that might benefit from AR-targeted therapy.

## 1. Introduction

The androgen receptor (AR) belongs to the steroid receptor subfamily of nuclear receptors [1], is a ligand-dependent nuclear transcription factor, and is encoded by the *AR* gene located on the short arm of the X chromosome (Xq11-12). AR binds androgens and these can exert their actions in a DNA binding-dependent manner to regulate target gene transcription [2] or in a non-DNA-binding dependent manner to initiate rapid, cellular events such as the phosphorylation of second messenger signaling cascades [3]. *AR* is widely expressed in various tissues with the highest levels reported for reproductive tissues (testes, prostate, ovaries, uterus) and has a diverse range of physiological functions, such as the development and maintenance of the reproductive, musculoskeletal, cardiovascular, immune, neural, and hemopoietic systems [4].

As expected from its wide expression in normal tissue (proteinatlas.org, accessed on 28 December 2023), various different tumor types can express *AR* [5]. AR has been extensively investigated in “classic” hormone-dependent cancers such as prostate and breast cancer. However, data on AR immunostaining in tumors are highly variable, especially in supposedly hormone-independent cancers. For example, the reported AR positivity ranges 13–54% in urothelial carcinoma [6,7,8,9], 54–100% in salivary duct carcinoma [10,11,12,13,14], 22–78% in basal cell carcinoma of the skin [15,16,17,18], and 13–67% in oral squamous cell carcinoma [19,20,21,22]. The main relevance of *AR* expression in cancer comes from its role as a therapeutic target. Various drugs targeting the AR in several different ways are routinely used in prostate cancer patients (summarized in [23,24]). Successful targeting of the AR has also been reported in bladder cancer [25], salivary gland carcinoma [26], and breast cancer [27], where the AR is supposed to interact with estrogen receptor signaling [28]. A summary of published data on *AR* expression in human tumors is given in Figure 1.

Considering that AR immunostaining has a diagnostic, potential prognostic, and predictive role in other tumors apart from prostate cancer, a comprehensive and highly standardized study analyzing a large number of tumors from different tumor entities is needed. Therefore, *AR* expression was analyzed in more than 18,000 tumor tissue samples from 141 different tumor types and subtypes as well as 76 different non-neoplastic tissue types by immunohistochemistry (IHC) in a tissue microarray (TMA) format in this study.

## 2. Material and Methods

Tissue Microarrays (TMAs). Two sets of TMAs were used in this study. The first was a normal tissue TMA that was constructed from 8 samples from 8 different donors for each of the 76 different normal tissue types. The second was a set of cancer TMAs that were made from 18,234 primary tumors from 141 tumor types and subtypes, which were distributed across 62 TMA blocks containing between 85 and 613 tissue spots per block. Histopathological data including grade, pT, and pN status were available from 1073 urothelial carcinomas, 1680 breast carcinomas, 182 endometrioid endometrial cancers, 1224 clear cell renal cell carcinomas, and 369 serous ovarian carcinomas.

For the tumor TMAs, a database with clinical follow-up data was available. It included data from 254 patients (median follow-up time 14 months, range 1–77 months) with urothelial carcinoma, from 717 patients (median follow-up time 50 months, range 1–88 months) with invasive breast carcinoma (NST), and from 531 patients (median follow-up time 40 months, 1–250 months) with clear cell renal cell carcinoma. A detailed description of the composition of the normal tissue TMA and the cancer TMAs is given in the results section. All samples were taken from the tissue archive of the Institute of Pathology, University Hospital of Hamburg, Germany, the Institute of Pathology, Clinical Center Osnabrueck, Germany, and the Department of Pathology, Academic Hospital Fuerth, Germany. All tissues had been fixed in 4% buffered formalin prior to paraffin embedding. A single 0.6 mm tissue spot per tumor was used for TMA construction. The use of leftover tissue samples from diagnostic tissues for TMA construction and of anonymized patient data for statistical analysis is in accordance with local laws (HmbKHG, §12) and has been approved by the local ethics committee (Ethics commission Hamburg, WF-049/09). All work has been carried out in compliance with the Helsinki Declaration.

Immunohistochemistry (IHC). All 63 TMA sections were freshly prepared and immunostained at the same day in one experiment. After paraffin removal, slides were rehydrated through a graded alcohol series and exposed to heat-induced antigen retrieval for 5 min in an autoclave at 121 °C in pH 7.8 buffer. Blocking of endogenous peroxidase with Dako Peroxidase Blocking Solution™ (Agilent Technologies Inc., Santa Clara, CA, USA; #S2023) was performed for 10 min. Primary antibody specific against AR (recombinant rabbit monoclonal, MSVA-367R; MS Validated Antibodies GmbH, Hamburg, Germany; #2145-376R) was applied at 37 °C for 60 min at a dilution of 1:450. The EnVision Kit™ (Agilent Technologies Inc., Santa Clara, CA, USA; #K5007) was used to detect bound antibody, and sections were counterstained with haemalaun. For the purpose of antibody validation, immunohistochemical staining of the normal tissue TMA was performed with a different anti-AR antibody (monoclonal rabbit, EPR1535(2), Abcam, Cambridge, UK; #133273) at a dilution of 1:300 and an otherwise identical protocol. One pathologist (SM) scored all tumors. For tumor tissues, the percentage of AR positive tumor cells was estimated and the staining intensity was semi-quantitatively recorded (0, 1+, 2+, 3+). The staining results were converted into four groups for statistical analysis: negative: no detectable staining; weak: staining intensity of 1+ in ≤70% or staining intensity of 2+ in ≤30% of tumor cells; moderate: staining intensity of 1+ in >70%, staining intensity of 2+ in >30% but in ≤70%, or staining intensity of 3+ in ≤30% of tumor cells; strong: staining intensity of 2+ in >70% or staining intensity of 3+ in >30% of tumor cells.

Statistics. Statistical calculations were performed with JMP 16 software (SAS Institute Inc., Cary, NC, USA). Contingency tables and the chi^2^-test were performed to search for associations between different AR staining levels, tumor phenotype, and patient gender. Survival curves for different levels of AR immunostaining in breast cancer samples were calculated according to Kaplan–Meier. The Log-Rank test was applied to detect significant differences between groups. A *p*-value of ≤0.05 was defined as significant.

## 3. Results

### 3.1. Technical Issues

A total of 14,408 (79.0%) of the arrayed tumors were interpretable for AR staining. Reasons for non-interpretable samples included unequivocal tumor cells or loss of the tissue spot during technical procedures. For each normal tissue type, a sufficient number of samples was evaluable.

### 3.2. AR Immunostaining in Normal Tissues

AR immunostaining was seen in a broad range of cell types and in most organs. AR staining was particularly strong in epithelial and stromal cells of the prostate and the seminal vesicle, tall columnar cells of the epididymis, Sertoli and Leydig cells of the testis, and in a subset of epithelial cells of the fallopian tube. A moderate nuclear AR staining was observed in sebaceous glands, stromal and epithelial cells of the endocervix, stroma cells of the endometrium, granulosa, theca interna, and stromal cells of the ovary, hepatocytes, a subset of glandular cells of the breast, basal cells of the respiratory epithelium, myometrium, and in skeletal muscle cells. In squamous epithelium, a weak to moderate staining of a fraction of cells occurred, predominantly in the lower half of the epithelium. A weak AR immunostaining also occurred in some cells of proximal tubuli and glomeruli in some kidney samples, gallbladder epithelium, excretory and intercalated ducts as well as some islet cells of the pancreas, glandular cells of salivary glands, gastric glands, thyroid glandular cells, urothelium, decidua cells of the pregnant uterus, smooth muscle cells, and in a fraction of cells in the aortic wall. AR immunostaining was completely absent in the adrenal gland, parathyroid, hematopoietic and lymphoid tissues, lung, hypophysis, and the brain. Representative images of AR staining are shown in Figure 2. All these findings were observed by both MSVA-367R and EPR1535(2) (Appendix A).

### 3.3. AR Immunostaining in Neoplastic Tissues

AR immunostaining was found in 30.8% of 14,408 cases, including 37.3% with weak, 16.1% with moderate, and 46.6% with strong positivity. A total of 116 (82.3%) of 141 tumor categories included at least one AR positive case and 66 (46.8%) contained at least one tumor with strong AR staining (Table 1; a graphical summary is given in Appendix A). Representative images of AR positive tumors are shown in Figure 3. A ranking of tumor categories according to the rate of AR positivity is given in Figure 4. A particularly high percentage of tumors with moderate to strong AR immunostaining was detected in sex cord-stromal tumors of the testis (93.3–100%), adenocarcinoma of the prostate (79.3–98.7%), breast neoplasms (25.0–75.5%), other gynecological tumors (0.9–100%), renal cell carcinomas (5.0–44.1%), and in urothelial carcinomas (5.4–24.2%).

### 3.4. AR Immunostaining, Tumor Phenotype, and Prognosis

Low AR staining was significantly associated with adverse histopathological and clinical features in several tumor types (Table 2). Low AR was linked to advanced tumor stage (pTa versus pT2-4; *p* < 0.0001) in urothelial carcinoma. In invasive breast carcinoma (NST), low AR staining was associated with high tumor stage (*p* < 0.0001), high tumor grade (*p* < 0.0001), lymph node metastasis (*p* < 0.0001), and shorter overall survival (*p* = 0.0094, Appendix A). In clear cell renal cell carcinoma (RCC), low AR staining was liked to high pT stage (*p* < 0.0001) and high tumor grade (*p* < 0.0001). In papillary RCC, low AR staining was related to high pT stage (*p* = 0.0055) as well as high ISUP (*p* = 0.0179) and Fuhrman grade (*p* = 0.0018). AR staining was unrelated to histopathological features and clinical features in 157 endometroid endometrial carcinomas and in 221 serous ovarian carcinomas.

### 3.5. AR Immunostaining, Gender and Age Distribution

Within non-mammary, non-gynecological, non-prostate, and non-testicular tumors, AR positivity was more common in tumors from male (25.9% of 4329) than from female patients (13.1% of 2868; *p* < 0.0001). This also held true within the separate cohorts of tumors of the urinary bladder and tumors of the kidney (*p* < 0.0001; Table 3). A subdivision of tumors from male patients in different age groups showed that AR positive tumors were significantly more common in older patients (*p* = 0.0044).

## 4. Discussion

More than 2650 articles listed in PubMed^®^ have described immunohistochemical evaluations of AR in cancer (PubMed^®^ search (10/23: “androgen receptor cancer immunohisto*”). The articles generally agree on prostatic adenocarcinomas being the most commonly AR positive cancer entities but published data on *AR* expression in other tumor entities vary considerably. The high diversity of published data on *AR* expression in human tumors primarily reflects the range of different antibodies, laboratory protocols, interpretation criteria, and thresholds to define positive AR staining in these studies. Our highly standardized analysis of 18,234 tumors from 141 human tumor types and subtypes has resulted in a ranking order according to the prevalence of *AR* expression. This represents the key finding of our study. A compilation of data from previous studies (Figure 1) demonstrates that such information could not be extracted from the literature. Our data demonstrate that AR positivity is similarly frequent in testicular and ovarian sex-cord stroma tumors as in prostatic adenocarcinomas although the *AR* expression levels tend to be lower in sex-cord stroma tumors and that neoplasms of the breast, other carcinomas of the female genital tract, renal cell carcinomas, and urothelial carcinomas represent further tumor categories with frequent and often strong *AR* expression. Most importantly, our data also show that—often at lower frequency—*AR* expression can also be found in a broad variety of other tumor types. The occurrence of high-level *AR* expression in many different tumor entities is consistent with the RNA expression data summarized in the TCGA database (https://www.cancer.gov/tcga, accessed on 28 December 2023).

In diagnostic pathology, several applications of AR immunostaining have been proposed but are not strongly supported by our data. Based on the frequent high-level *AR* expression in prostatic adenocarcinoma, AR IHC is applied for the distinction of poorly differentiated prostatic cancer from urothelial carcinoma. For example, Downes et al. described intense AR positivity in 100% of prostatic cancers and negative or only weak staining in urothelial carcinomas and found that AR IHC was superior to prostate specific markers such as PSA and PSAP in discriminating poorly differentiated urothelial carcinoma from high grade prostate carcinoma [29]. However, considering that 27.7% of muscle-invasive urothelial carcinomas were AR positive (5.6% with strong AR staining) in our study, the diagnostic potential of AR appears to be limited in this setting. AR immunostaining was also proposed for the distinction of sebaceous from basal cell and squamous cell carcinoma of the skin [30]. Yet in our study, 64.2% of basal cell carcinomas and 11.5% of squamous cell carcinomas of the skin were AR positive. In breast cancer, AR IHC has been proposed as a tool for the detection of apocrine differentiation [31] and metastases from triple-negative cancers [32]. Given that in 54.7% of our breast carcinomas, NST showed strong AR positivity, AR IHC may not be optimal for verifying apocrine differentiation. Since only 36.8% of triple negative breast carcinomas showed AR positivity in this study and the overall specificity of AR for breast cancer was low, the use of AR IHC for verifying breast cancer origin of AR positive metastases is limited. With GATA3 and TRPS1, more specific breast cancer markers are available that showed 55.4% (GATA3, [33]) and 77.8% positivity (TRPS1, manuscript in revision) in triple-negative breast cancers in our tumor cohort.

The large number of tumors analyzed for several tumor categories enabled an analysis of the potential clinical significance of AR expression. In breast, urothelial, and renal cell carcinomas, detectable AR staining was associated with favorable histopathological and clinical features. For breast cancer, previous data are controversial. Some authors also reported a favorable prognostic role of *AR* expression [34,35], but others found no association between *AR* expression and prolonged disease-free survival [36]. The correlation between high *AR* expression and the non-invasive stage for urothelial carcinoma is consistent with results from several earlier studies [37,38,39]. The significant association between positive AR staining and favorable tumor features seen in our clear cell carcinomas of the kidney is also in line with previous studies [9,37,40,41,42]. Given the tendency towards adjuvant therapy in high-risk renal cell carcinomas and the corresponding need for risk assessment in these tumors, AR might deserve further evaluation as a potential clinically useful prognostic marker in these tumors.

Most of the importance of AR comes from its well-proven role as a highly suited therapeutic target for adenocarcinomas of the prostate [23]. Given the frequent and often high-level *AR* expression in many other tumor types, it is obvious to consider AR targeting therapies for these tumors. The successful use of anti-androgen drugs has been reported in various cancer types including carcinomas of the breast [43], salivary gland carcinoma [44,45], ovarian cancer [46], bladder cancer [47], and salivary duct carcinoma [10,48,49]. Clinical trials on the use of anti-androgen drugs in non-prostate cancers are currently underway (according to clinicaltrials.gov, accessed on 28 December 2023) in breast carcinoma (especially triple negative breast cancer) (n = 12), salivary gland carcinoma (n = 6), or bladder cancer (n = 2). Once a more widespread clinical utility of these drugs should become evident, our ranking order of tumors according to their AR positivity rates would define the tumor entities that can benefit most from such approaches and which would be best suited for further clinical trials.

Whether AR has a biological role in tumors with only minor immunohistochemical AR positivity remains unclear. Mechanistically, AR associates with heat shock proteins and cellular chaperones in the cytoplasm in its inactive form. Upon binding of androgens, AR dissociates from the complex, forms homodimers, and migrates into the nucleus. Such activated AR binds to androgen response elements together with co-regulators and leads to the transcriptional regulation of a large variety of target genes. However, the number and kind of AR-co-regulators and AR-responsive genes in different normal and cancerous tissues, particularly in cancers other than prostatic tumors, and the impact of the level of AR are still the subjects of current research [50,51,52]. The theory that low-level *AR* expression is non-random is supported by the significantly higher rate of positive AR immunostaining in “non-genital” tumors from male than from female patients. Since comprehensive understanding of the mechanisms controlling the expression of the receptor in different target tissues is lacking, it can only be hypothesized that this is due to gender-associated differences in the testosterone serum levels. Intuitively, one would assume that a cell would benefit more from AR upregulation in a testosterone-rich environment. It should be noted, however, that regulation of AR is cell- and tissue specific and androgens can have a negative as well as positive auto-regulation on *AR* expression [53]. A negative auto-regulation by dihydrotestosterone on *AR* expression [53] could potentially explain the lower percentage of AR-positive tumors in younger male patients.

The large scale of our study prompted us to place special emphasis on a thorough validation of our assay. We followed the proposal of the International Working Group for Antibody Validation (IWGAV). According to the IWGAV, validation of antibodies used for IHC on formalin-fixed tissues must include either a comparison of the findings obtained by two different independent antibodies or a comparison with expression data obtained by another independent method [54]. The rather ubiquitous expression of AR identified in our IHC analysis of normal tissues is largely consistent with RNA expression data from three public databases [55], which were compiled in the Human Protein Atlas [56,57]. but given the cell type specificity of AR expression, a comparison with a method based on disaggregated tissue is suboptimal for this protein. The critical evidence for the validity of our assay comes from confirmation that all AR positive cell types seen by MSVA-367R were also seen by the independent second antibody EPR1535(2). It is of note that the use of a very broad range of different tissues (76 different normal tissue categories) for antibody validation increases the likelihood of detecting undesired cross-reactivities because virtually all proteins occurring in normal cells of adult humans are subjected to the validation experiment.

There were some limitations to our study. The number of tumors analyzed for many common tumor types was not always as high as would have been desirable, and some rare tumor types were not even represented in our series of TMAs. In addition, clinical follow-up data were not available for all samples and tumor types. It would be interesting for future studies to investigate the clinical significance of *AR* expression in all these different tumor types that recurrently showed positive AR immunostaining.

## 5. Conclusions

Our analysis of 141 different tumor types for AR immunostaining provides a comprehensive overview on AR in human tumors and identifies *AR* expression in many “hormone-dependent” and “hormone-independent” tumor types. The data suggest a limited role of AR IHC for the distinction of tumors and a strong prognostic role in breast cancer and clear cell RCC and highlight numerous tumor entities that could benefit from androgen-targeting cancer drugs.

## Figures and Tables

**Figure 1 biomedicines-12-00957-f001:**
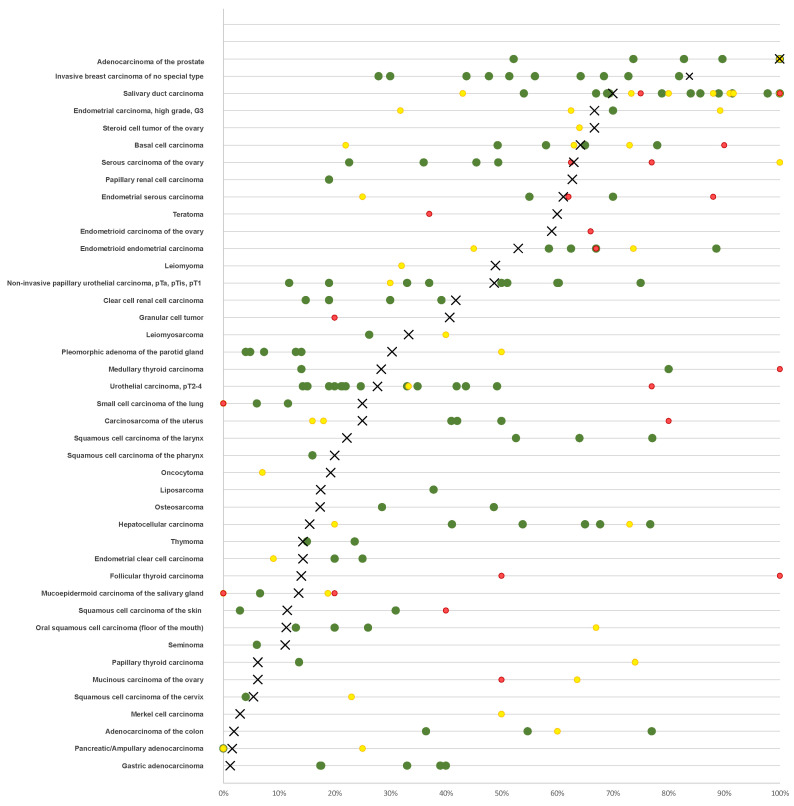
Comparison with previous AR literature. An “X” indicates the fraction of AR positive cancers in the present study, dots indicate the reported frequencies from the literature for comparison: red dots mark studies with <10 analyzed tumors, yellow dots mark studies with >10–25 analyzed tumors, and green dots mark studies with >25 analyzed tumors.

**Figure 2 biomedicines-12-00957-f002:**
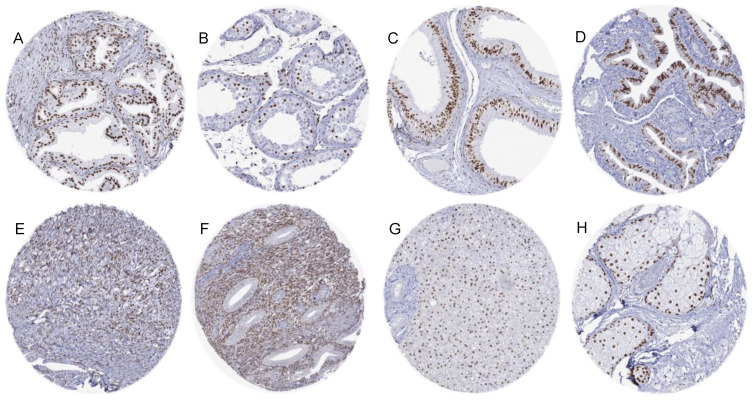
AR immunostaining in normal tissues. (**A**) Positive AR immunostaining of epithelial cells, basal cells, and stromal cells in the prostate gland. (**B**) Positive AR immunostaining of Sertoli cells and Leydig cells in testis. (**C**) Positive AR immunostaining of epithelial and stromal cells in epididymis. (**D**) Positive AR immunostaining of secretory cells in the fallopian tube. (**E**) Positive AR immunostaining of ovarian stromal cells (**F**) Positive AR immunostaining of stromal cells in proliferative endometrium. (**G**) Positive AR immunostaining of hepatocytes. (**H**) Positive AR immunostaining in a sebaceous gland of the skin.

**Figure 3 biomedicines-12-00957-f003:**
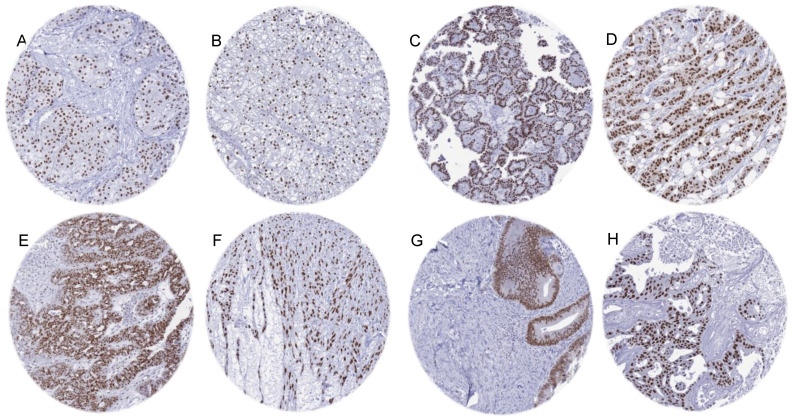
Moderate to strong AR immunostaining in “non-genital” tumors. (**A**) Urothelial carcinoma of the bladder. (**B**) Clear cell renal cell carcinoma. (**C**) Papillary renal cell carcinoma. (**D**) Invasive breast carcinoma of no special type (NST). (**E**) High-grade serous carcinoma of the ovary. (**F**) Uterine leiomyosarcoma. (**G**) Adenocarcinoma of the colon. (**H**) Salivary duct carcinoma.

**Figure 4 biomedicines-12-00957-f004:**
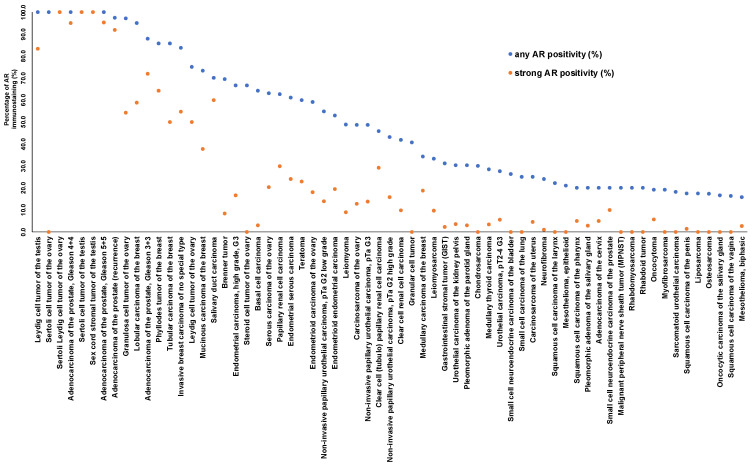
Ranking order of AR immunostaining in cancers. Both the frequency of positive cases (blue dots) and the frequency of strongly positive cases (orange dots).

**Table 1 biomedicines-12-00957-t001:** AR immunostaining in tumors. Int. = interpretable, Neg. = negative, Mod. = moderate, Str. = strong.

	Tumor Entity		AR Immunostaining				
		On TMA (n)	Int. (n)	Neg. (%)	Weak (%)	Mod. (%)	Str. (%)
Tumors of the Skin	Pilomatricoma	35	22	86.4	13.6	0.0	0.0
	Basal cell carcinoma	88	67	35.8	50.7	10.4	3.0
	Benign nevus	29	18	100.0	0.0	0.0	0.0
	Squamous cell carcinoma of the skin	90	78	88.5	10.3	1.3	0.0
	Malignant melanoma	46	38	100.0	0.0	0.0	0.0
	Malignant melanoma lymph node metastasis	86	84	97.6	2.4	0.0	0.0
	Merkel cell carcinoma	46	33	97.0	3.0	0.0	0.0
Tumors of the head and neck	Squamous cell carcinoma of the larynx	109	81	77.8	22.2	0.0	0.0
	Squamous cell carcinoma of the pharynx	60	60	80.0	13.3	1.7	5.0
	Oral squamous cell carcinoma (floor of the mouth)	130	115	88.7	9.6	0.9	0.9
	Pleomorphic adenoma of the parotid gland	50	33	69.7	27.3	0.0	3.0
	Warthin tumor of the parotid gland	104	75	100.0	0.0	0.0	0.0
	Adenocarcinoma, NOS (papillary cystadenocarcinoma)	14	8	100.0	0.0	0.0	0.0
	Salivary duct carcinoma	15	10	30.0	0.0	10.0	60.0
	Acinic cell carcinoma of the salivary gland	181	100	97.0	1.0	1.0	1.0
	Adenocarcinoma NOS of the salivary gland	109	58	87.9	1.7	3.4	6.9
	Adenoid cystic carcinoma of the salivary gland	180	99	94.9	5.1	0.0	0.0
	Basal cell adenocarcinoma of the salivary gland	25	15	93.3	0.0	0.0	6.7
	Basal cell adenoma of the salivary gland	101	62	87.1	12.9	0.0	0.0
	Epithelial–myoepithelial carcinoma of the salivary gland	53	42	88.1	11.9	0.0	0.0
	Mucoepidermoid carcinoma of the salivary gland	343	281	86.5	2.8	1.4	9.3
	Myoepithelial carcinoma of the salivary gland	21	16	93.8	6.3	0.0	0.0
	Myoepithelioma of the salivary gland	11	9	88.9	11.1	0.0	0.0
	Oncocytic carcinoma of the salivary gland	12	6	83.3	16.7	0.0	0.0
	Polymorphous adenocarcinoma, low grade, of the salivary gland	41	21	100.0	0.0	0.0	0.0
	Pleomorphic adenoma of the salivary gland	53	35	80.0	14.3	2.9	2.9
Tumors of the lung, pleura, and thymus	Adenocarcinoma of the lung	196	100	96.0	2.0	2.0	0.0
	Squamous cell carcinoma of the lung	80	37	86.5	10.8	2.7	0.0
	Small cell carcinoma of the lung	16	8	75.0	12.5	12.5	0.0
	Mesothelioma, epithelioid	39	19	78.9	21.1	0.0	0.0
	Mesothelioma, biphasic	76	38	84.2	13.2	0.0	2.6
	Thymoma	29	21	85.7	9.5	0.0	4.8
	Lung, neuroendocrine tumor (NET)	19	16	87.5	6.3	6.3	0.0
Tumors of the female genital tract	Squamous cell carcinoma of the vagina	78	49	83.7	12.2	4.1	0.0
	Squamous cell carcinoma of the vulva	130	115	87.8	10.4	0.0	1.7
	Squamous cell carcinoma of the cervix	128	112	94.6	4.5	0.9	0.0
	Adenocarcinoma of the cervix	21	20	80.0	15.0	0.0	5.0
	Endometrioid endometrial carcinoma	236	200	47.0	21.0	12.5	19.5
	Endometrial serous carcinoma	82	54	38.9	29.6	7.4	24.1
	Carcinosarcoma of the uterus	48	44	75.0	15.9	4.5	4.5
	Endometrial carcinoma, high grade, G3	13	12	33.3	41.7	8.3	16.7
	Endometrial clear cell carcinoma	8	7	85.7	0.0	0.0	14.3
	Endometrioid carcinoma of the ovary	110	83	41.0	28.9	12.0	18.1
	Serous carcinoma of the ovary	559	338	37.0	29.9	12.7	20.4
	Mucinous carcinoma of the ovary	96	65	93.8	3.1	0.0	3.1
	Clear cell carcinoma of the ovary	50	37	97.3	0.0	2.7	0.0
	Carcinosarcoma of the ovary	47	39	51.3	25.6	10.3	12.8
	Granulosa cell tumor of the ovary	37	35	2.9	14.3	28.6	54.3
	Leydig cell tumor of the ovary	4	4	25.0	25.0	0.0	50.0
	Sertoli cell tumor of the ovary	1	1	0.0	0.0	100.0	0.0
	Sertoli Leydig cell tumor of the ovary	3	3	0.0	0.0	0.0	100.0
	Steroid cell tumor of the ovary	3	3	33.3	0.0	66.7	0.0
	Brenner tumor	41	36	30.6	44.4	16.7	8.3
Tumors of the breast	Invasive breast carcinoma of no special type	1764	1507	16.3	16.3	12.6	54.7
	Lobular carcinoma of the breast	363	277	5.1	19.5	16.6	58.8
	Medullary carcinoma of the breast	34	32	65.6	9.4	6.3	18.8
	Tubular carcinoma of the breast	29	14	14.3	21.4	14.3	50.0
	Mucinous carcinoma of the breast	65	45	26.7	20.0	15.6	37.8
	Phyllodes tumor of the breast	50	42	14.3	14.3	7.1	64.3
Tumors of the digestive system	Adenomatous polyp, low-grade dysplasia	50	41	95.1	4.9	0.0	0.0
	Adenomatous polyp, high-grade dysplasia	50	47	97.9	0.0	2.1	0.0
	Adenocarcinoma of the colon	2482	1927	98.1	1.5	0.2	0.2
	Gastric adenocarcinoma, diffuse type	176	148	100.0	0.0	0.0	0.0
	Gastric adenocarcinoma, intestinal type	174	165	98.8	0.0	0.6	0.6
	Gastric adenocarcinoma, mixed type	62	57	100.0	0.0	0.0	0.0
	Adenocarcinoma of the esophagus	83	76	100.0	0.0	0.0	0.0
	Squamous cell carcinoma of the esophagus	76	70	97.1	0.0	1.4	1.4
	Squamous cell carcinoma of the anal canal	89	74	89.2	8.1	1.4	1.4
	Cholangiocarcinoma	113	95	96.8	2.1	1.1	0.0
	Gallbladder adenocarcinoma	31	29	96.6	3.4	0.0	0.0
	Gallbladder Klatskin tumor	41	33	100.0	0.0	0.0	0.0
	Hepatocellular carcinoma	300	284	84.5	9.9	2.8	2.8
	Ductal adenocarcinoma of the pancreas	612	376	99.2	0.8	0.0	0.0
	Pancreatic/ampullary adenocarcinoma	89	62	98.4	1.6	0.0	0.0
	Acinar cell carcinoma of the pancreas	16	16	93.8	6.3	0.0	0.0
	Gastrointestinal stromal tumor (GIST)	50	45	68.9	28.9	0.0	2.2
	Appendix, neuroendocrine tumor (NET)	22	9	88.9	11.1	0.0	0.0
	Colorectal, neuroendocrine tumor (NET)	12	5	100.0	0.0	0.0	0.0
	Ileum, neuroendocrine tumor (NET)	49	36	100.0	0.0	0.0	0.0
	Pancreas, neuroendocrine tumor (NET)	97	87	92.0	6.9	1.1	0.0
	Colorectal, neuroendocrine carcinoma (NEC)	12	5	100.0	0.0	0.0	0.0
	Gallbladder, neuroendocrine carcinoma (NEC)	4	2	100.0	0.0	0.0	0.0
	Pancreas, neuroendocrine carcinoma (NEC)	14	14	92.9	7.1	0.0	0.0
Tumors of the urinary system	Non-invasive papillary urothelial carcinoma, pTa G2 low grade	177	115	45.2	32.2	8.7	13.9
	Non-invasive papillary urothelial carcinoma, pTa G2 high grade	141	95	56.8	18.9	8.4	15.8
	Non-invasive papillary urothelial carcinoma, pTa G3	219	152	51.3	28.3	6.6	13.8
	Urothelial carcinoma, pT2-4 G3	735	575	72.3	17.0	5.0	5.6
	Squamous cell carcinoma of the bladder	22	20	100.0	0.0	0.0	0.0
	Small cell neuroendocrine carcinoma of the bladder	23	19	73.7	15.8	10.5	0.0
	Sarcomatoid urothelial carcinoma	25	11	81.8	18.2	0.0	0.0
	Urothelial carcinoma of the kidney pelvis	62	56	69.6	25.0	1.8	3.6
	Clear cell renal cell carcinoma	1287	1186	58.2	20.8	11.1	9.9
	Papillary renal cell carcinoma	368	338	37.3	18.6	14.2	29.9
	Clear cell (tubulo) papillary renal cell carcinoma	26	24	54.2	8.3	8.3	29.2
	Chromophobe renal cell carcinoma	170	160	84.4	10.6	1.9	3.1
	Oncocytoma	257	244	80.7	9.4	4.1	5.7
Tumors of the male genital organs	Adenocarcinoma of the prostate, Gleason 3 + 3	83	82	12.2	8.5	7.3	72.0
	Adenocarcinoma of the prostate, Gleason 4 + 4	80	79	0.0	1.3	3.8	94.9
	Adenocarcinoma of the prostate, Gleason 5 + 5	85	84	0.0	3.6	1.2	95.2
	Adenocarcinoma of the prostate (recurrence)	258	196	2.6	3.1	2.6	91.8
	Small cell neuroendocrine carcinoma of the prostate	19	10	80.0	10.0	0.0	10.0
	Seminoma	621	595	88.9	10.4	0.7	0.0
	Embryonal carcinoma of the testis	50	44	97.7	2.3	0.0	0.0
	Leydig cell tumor of the testis	30	30	0.0	6.7	10.0	83.3
	Sertoli cell tumor of the testis	2	2	0.0	0.0	0.0	100.0
	Sex cord stromal tumor of the testis	1	1	0.0	0.0	0.0	100.0
	Spermatocytic tumor of the testis	1	1	100.0	0.0	0.0	0.0
	Yolk sac tumor	50	37	91.9	8.1	0.0	0.0
	Teratoma	50	35	40.0	25.7	11.4	22.9
	Squamous cell carcinoma of the penis	80	74	82.4	12.2	4.1	1.4
Tumors of endocrine organs	Adenoma of the thyroid gland	113	96	85.4	14.6	0.0	0.0
	Papillary thyroid carcinoma	391	241	93.8	5.4	0.0	0.8
	Follicular thyroid carcinoma	154	114	86.0	8.8	2.6	2.6
	Medullary thyroid carcinoma	111	88	71.6	22.7	2.3	3.4
	Parathyroid gland adenoma	43	42	97.6	2.4	0.0	0.0
	Anaplastic thyroid carcinoma	45	39	84.6	10.3	5.1	0.0
	Adrenal cortical adenoma	50	48	100.0	0.0	0.0	0.0
	Adrenal cortical carcinoma	26	24	100.0	0.0	0.0	0.0
	Pheochromocytoma	50	48	100.0	0.0	0.0	0.0
Lymphoma	Hodgkin’s lymphoma	45	27	100.0	0.0	0.0	0.0
Tumors of soft tissue and bone	Tendosynovial giant cell tumor	45	25	100.0	0.0	0.0	0.0
	Granular cell tumor	53	27	59.3	25.9	14.8	0.0
	Leiomyoma	50	45	51.1	26.7	13.3	8.9
	Leiomyosarcoma	87	72	66.7	20.8	2.8	9.7
	Liposarcoma	132	80	82.5	15.0	2.5	0.0
	Malignant peripheral nerve sheath tumor (MPNST)	13	10	80.0	20.0	0.0	0.0
	Myofibrosarcoma	26	26	80.8	15.4	3.8	0.0
	Angiosarcoma	73	43	100.0	0.0	0.0	0.0
	Angiomyolipoma	91	60	85.0	11.7	0.0	3.3
	Dermatofibrosarcoma protuberans	21	12	100.0	0.0	0.0	0.0
	Ganglioneuroma	14	14	100.0	0.0	0.0	0.0
	Neurofibroma	117	104	76.0	23.1	0.0	1.0
	Sarcoma, not otherwise specified (NOS)	74	62	85.5	12.9	1.6	0.0
	Paraganglioma	41	41	97.6	2.4	0.0	0.0
	Ewing sarcoma	23	8	100.0	0.0	0.0	0.0
	Rhabdomyosarcoma	6	5	80.0	0.0	20.0	0.0
	Schwannoma	121	114	98.2	1.8	0.0	0.0
	Synovial sarcoma	12	8	100.0	0.0	0.0	0.0
	Osteosarcoma	43	23	82.6	17.4	0.0	0.0
	Chondrosarcoma	38	10	70.0	30.0	0.0	0.0
	Rhabdoid tumor	5	5	80.0	20.0	0.0	0.0

**Table 2 biomedicines-12-00957-t002:** AR immunostaining and prognosis in invasive breast carcinoma (NST), urothelial carcinoma, and clear cell carcinoma of the kidney. Neg. = negative, Mod. = moderate, Str. = strong.

			AR IHC Result					*p*
			n	Neg. (%)	Weak (%)	Mod. (%)	Str. (%)	
Breast cancer of no special type	Tumor stage	pT1	688	11.5	13.5	12.1	62.9	<0.0001
		pT2	582	18.0	18.2	12.9	50.9	
		pT3-4	120	26.7	23.3	9.2	40.8	
	Grade	G1	169	5.3	11.8	8.9	74.0	<0.0001
		G2	746	10.2	17.3	12.9	59.7	
		G3	512	27.1	17.2	12.3	43.4	
	Nodal stage	pN0	635	12.4	15.6	12.9	59.1	<0.0001
		pN1	302	16.2	16.9	13.2	53.6	
		pN2	112	20.5	22.3	14.3	42.9	
		pN3	64	23.4	34.4	14.1	28.1	
	HER2 status	Negative	795	15.6	13.7	10.1	60.6	0.0050
		Positive	112	8.0	15.2	20.5	56.3	
	ER status	Negative	188	46.8	17.6	10.6	25.0	<0.0001
		Positive	680	6.3	12.2	11.9	69.6	
	PR status	Negative	367	30.2	17.2	13.1	39.5	<0.0001
		Positive	535	4.7	11.0	10.7	73.6	
	Triple negative	No	718	7.0	13.1	12.7	67.3	<0.0001
		Yes	125	63.2	16.8	4.8	15.2	
Urinary bladder cancer	Tumor stage	pTa G2 low	115	45.2	32.2	8.7	13.9	<0.0001
		pTa G2 high	95	56.8	18.9	8.4	15.8	
		pTa G3	122	51.6	26.2	5.7	16.4	
		pT2	122	69.7	19.7	4.1	6.6	0.9943
		pT3	207	72.9	17.9	3.9	5.3	
		pT4	96	74.0	16.7	4.2	5.2	
	Nodal stage	pN0	248	78.6	14.1	3.2	4.0	0.0052
		pN+	170	62.3	24.7	4.5	8.4	
Clear cell renal cell cancers	ISUP stage	1	262	53.8	25.6	11.5	9.2	<0.0001
		2	398	54.3	18.6	15.6	11.6	
		3	262	64.9	21.4	7.3	6.5	
		4	77	77.9	13.0	7.8	1.3	
	Fuhrman grade	G1	65	40	32.3	16.9	10.8	<0.0001
		G2	672	54	20.7	12.8	12.5	
		G3	293	65.9	21.2	7.2	5.8	
		G4	92	76.1	12.0	9.8	2.2	
	Thoenes grade	G1	349	52.7	23.8	11.5	12.0	<0.0001
		G2	484	63.8	18.6	10.1	7.4	
		G3	101	81.2	12.9	4.0	2.0	
	UICC stage	1	336	49.4	25.0	14.6	11.0	<0.0001
		2	37	83.8	10.8	2.7	2.7	
		3	90	75.6	16.7	4.4	3.3	
		4	73	84.9	13.7	1.4	0.0	
	pT stage	pT1	677	46.2	23.9	15.7	14.2	<0.0001
		pT2	127	78.7	15.0	3.1	3.1	
		pT3-4	323	74.9	15.8	5.9	3.4	
	Nodal stage	pN0	168	71.4	13.7	8.3	6.5	0.3279
		pN ≥ 1	27	85.2	3.7	7.4	3.7	
	Distant mets stage	pM0	111	61.3	18.0	12.6	8.1	0.0075
		pM ≥ 1	92	77.2	17.4	4.3	1.1	

**Table 3 biomedicines-12-00957-t003:** AR immunostaining in female and male patients (excluding tumors of the female genital tract, tumors of the male genital tract, and breast tumors).

Tumor Set	Sex	AR Neg. (%)	AR Weak (%)	AR Mod. (%)	AR Strong (%)	*p*
All tumors *	Female (n = 2868)	86.9	8.8	2.3	2	*p* < 0.0001
	Male (n = 4329)	74.1	13.7	5.2	7	
Tumors of the urinary bladder	Female (n = 474)	83.8	12.4	3.2	0.6	*p* < 0.0001
	Male (n = 1584)	74.1	18.8	5.3	1.9	
Tumors of the kidney	Female (n = 585)	72	17.1	7.4	3.6	*p* < 0.0001
	Male (n = 1262)	53.4	18.7	11.4	16.5	
Tumors of the colon	Female (n = 375)	98.7	1.3	0	0	*p* = 0.1085
	Male (n = 475)	96.8	2.7	0	0.4	

* Excluding tumors of the female genital tract, tumors of the male genital tract, and breast tumors. Neg. = negative.

## Data Availability

All data generated or analyzed during this study are included in this published article.

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
