# Peer review of "Frequency of Androgen Receptor Positivity in Tumors: A Study Evaluating More Than 18,000 Tumors"

_biomedicines, 2024, doi:10.3390/biomedicines12050957_

Round 1

Reviewer 1 Report

Comments and Suggestions for Authors

This is an interesting article, however, some points have to be fixed before acceptance.

1)      I strongly recommend drawing an appropriate graphical abstract at the end of the introduction.

2)      It was not clear that the whole of the methods has been done by the authors or some parts have been carried out by the authors and some parts have been compared with previous research.  Especially, when they claimed “in more than 18000 tumors have been studied”.

3)      How androgen receptor has a role in breast cancer ( line 20-21)

4)      It is good that the authors present tables 1 and 2 as graphs or columns. It is easier to get information

5)      It is good the authors pay attention to the mechanistic of why the androgen receptor has functions in various cancers in the Dissection part. 

Author Response

This is an interesting article, however, some points have to be fixed before acceptance.

  • I strongly recommend drawing an appropriate graphical abstract at the end of the introduction.

Reply: We have included a reference to the graphical summary of the existing literature on AR in human tumors at the end of the introduction on page 2, lines 58-59.

  • It was not clear that the whole of the methods has been done by the authors or some parts have been carried out by the authors and some parts have been compared with previous research.  Especially, when they claimed “in more than 18000 tumors have been studied”.

Reply: We have now better detailed that the >18,000 tumors were distributed across 62 TMA slides (page 3, line 77-79), and that one pathologists scored all >18,000 tumors (page 4, line 111).

  • How androgen receptor has a role in breast cancer ( line 20-21)

Reply: We have briefly outlined the role of AR in breast cancer in the introduction on page 2, lines 57-58.

  • It is good that the authors present tables 1 and 2 as graphs or columns. It is easier to get information.
  • It is good the authors pay attention to the mechanistic of why the androgen receptor has functions in various cancers in the Dissection part.

Reply zu 4-5: We thank the reviewer for his positive assessment.

Reviewer 2 Report

Comments and Suggestions for Authors

The article offers a thorough analysis of AR expression across diverse tumor types and normal tissues, bolstering its credibility through a substantial sample size and the identification of AR positivity in numerous tumor types. Nonetheless, the article have some areas needing improvement.

1. The organization of references in the article appears disjointed, warranting a need for sequential arrangement of citations.

2.  The variability observed in AR immunostaining data within tumors raises pertinent questions regarding the practical implications of research in this field. It is advisable for the authors to elucidate the underlying reasons or propose hypotheses to explain this variability. Without such discussion or hypothesis elucidation, the integrity of the study's rationale may be compromised.

3. In statistical description, it is advisable for authors to provide a more detailed description of the statistical methods employed during the experimental process, including specific parameters utilized.

4. The manuscript frequently employs a mix of "Androgen receptor" and "AR" terminology, necessitating rectification by the authors for consistency and clarity.

5.In the discussion section, it is advisable for the authors to provide a comprehensive overview of the limitations of the study and suggest potential avenues for future research. 

6.The manuscript contains numerous descriptions that overlap with previously published work, indicating a need for the authors to consolidate and refine their presentation to avoid redundancy and contribute novel insights to the field.

7.The authors have not clearly summarized the novelty and contributions of their research. It is recommended that they provide a summary of key findings and their significance to effectively highlight the innovative aspects and contributions of the study.

8.There are some references in the bibliography that have not been cited in the main text of the manuscript.

Author Response

The article offers a thorough analysis of AR expression across diverse tumor types and normal tissues, bolstering its credibility through a substantial sample size and the identification of AR positivity in numerous tumor types. Nonetheless, the article have some areas needing improvement.

The organization of references in the article appears disjointed, warranting a need for sequential arrangement of citations.

Reply: We have re-built the reference list with sequential arrangement of the citations.

The variability observed in AR immunostaining data within tumors raises pertinent questions regarding the practical implications of research in this field. It is advisable for the authors to elucidate the underlying reasons or propose hypotheses to explain this variability. Without such discussion or hypothesis elucidation, the integrity of the study's rationale may be compromised.

Reply: We have now explained the reasons for the variability of literature data on page 15, lines 229-235.

In statistical description, it is advisable for authors to provide a more detailed description of the statistical methods employed during the experimental process, including specific parameters utilized.

Reply: We have now better described the statistical tests used in the M&M section on page 4 lines 121-124.

The manuscript frequently employs a mix of "Androgen receptor" and "AR" terminology, necessitating rectification by the authors for consistency and clarity.

Reply: We now introduce the term “androgen receptor (AR)” at first occurrence in the abstract and in the article body and then use the abbreviation “AR” throughout the manuscript.

In the discussion section, it is advisable for the authors to provide a comprehensive overview of the limitations of the study and suggest potential avenues for future research.

Reply: We habe now dicussed the limitations and potentil future research of the study on page 18, lines 328-333.

The manuscript contains numerous descriptions that overlap with previously published work, indicating a need for the authors to consolidate and refine their presentation to avoid redundancy and contribute novel insights to the field.

Reply: We revised our manuscript and re-phrased the respective paragraphs on page 2-3 (Materials and Methods, Tissue microarrays, immunohistochemistry) to minimize overlap with our previous work.

The authors have not clearly summarized the novelty and contributions of their research. It is recommended that they provide a summary of key findings and their significance to effectively highlight the innovative aspects and contributions of the study.

Reply: We now better explained the novelty and impact of our findings on page 15, lines 229-235.

There are some references in the bibliography that have not been cited in the main text of the manuscript.

Reply: We removed the unused references.

Round 2

Reviewer 1 Report

Comments and Suggestions for Authors

The Authors did not answer some part of my questions and then I have to reject it. 

Author Response

  • It is good that the authors present tables 1 and 2 as graphs or columns. It is easier to get information.

Reply: We now provide a graphical version of the data of table in Supplementary Figure 2. We would like to keep that definite data (percentages) in table 1, however. We do not provide a graphical version of table 2, however, because percentages can be shown more precisely in a tabular format.

  • It is good the authors pay attention to the mechanistic of why the androgen receptor has functions in various cancers in the Dissection part.

Reply: We have now better explained the mechanistics of AR function in the discussion on page 17, lines 298-305.

Reviewer 2 Report

Comments and Suggestions for Authors

In this revision, the journal did not provide a plagiarism report. However, based on the previous plagiarism report and the revisions made in this iteration, there are still significant textual overlap with prior works. The author is advised to make corresponding adjustments accordingly.

Author Response

In this revision, the journal did not provide a plagiarism report. However, based on the previous plagiarism report and the revisions made in this iteration, there are still significant textual overlap with prior works. The author is advised to make corresponding adjustments accordingly.

Reply: We agree that there is considerable textual overlap with our previous publications. However, this is almost completely limited to the Materials and Methods section, and due to the fact that we use an identical or at least very similar set of tissues (TMAs), highly standardized laboratory protocols, standardized scoring criteria, and standard ethical declaration. We have now substantially rephrased these sections wherever possible.

Round 3

Reviewer 1 Report

Comments and Suggestions for Authors

The authors did not provide a graphical abstract at the end of the introduction. The subject of this paper is tough and the graphical abstract is required.

2) Moreover, the authors provided a figure representing published data ( papers) on AR expression in human tumors.  This is a research article and this figure is more suitable for the review articles.  They could write a sentence regarding the importance of AR in human tumors.